# Study on the Optimum Steel Slag Content of SMA-13 Asphalt Mixes Based on Road Performance

**Wei Chen [1], Jincheng Wei [1,2,*], Xizhong Xu [2], Xiaomeng Zhang [2], Wenyang Han [2], Xiangpeng Yan [2], Guiling Hu [1] and Zizhao Lu [3]**

1   School of Transportation Engineering, Shandong Jianzhu University, Jinan 250101, China; chenwei13572468@163.com (W.C.); huguilingtech@Foxmail.com (G.H.)
2   Science and Technology Innovation Center, Shandong Transportation Institute, Jinan 250102, China; xxz137152@163.com (X.X.); zhangxiaomeng@sdjtky.cn (X.Z.); hanwenyang@sdjtky.cn (W.H.); yanxiangpeng336@163.com (X.Y.)
3   Shandong Transportation Planning and Design Research Institute Co., Ltd., Jinan 250031, China; sdjtlzz@163.com
*   Correspondence: c2523860697@163.com

**Abstract:** To reduce the use of aggregates such as limestone and basalt, this paper used steel slag to replace some of the limestone aggregates in the production of SMA-13 asphalt mixes. The optimum content of steel slag in the SMA-13 asphalt mixes was investigated, and the performance of these mixes was evaluated. Five SMA-13 asphalt mixes with varying steel slag content (0%, 25%, 50%, 75%, and 100%) were designed and prepared experimentally. The high-temperature stability, low-temperature crack resistance, water stability, dynamic modulus, shear resistance, and volumetric stability of the mixes were investigated using the wheel tracking, Hamburg wheel tracking, three-point bending, freeze–thaw splitting, dynamic modulus, uniaxial penetration, and asphalt mix expansion tests. The results showed that compared to normal SMA-13 asphalt mixes, the high-temperature stability, water stability, and shear resistance of the SMA-13 asphalt mixes increased and then decreased as the steel slag content increased. All three performance indicators peaked at 75% steel slag content, and the dynamic stability, freeze–thaw splitting ratio, and uniaxial penetration strength increased by 90.48%, 7.39%, and 88.08%, respectively; however, the maximum bending tensile strain, which represents the low-temperature crack resistance of the asphalt mix, decreased by 5.98%. The dynamic modulus of the SMA-13 asphalt mixes increased with increasing steel slag content, but the volume expansion at a 75% steel slag content was 0.446% higher than at a 0% steel slag content. Based on the experimental results, the optimum content of steel slag for SMA-13 asphalt mixes was determined to be 75%.

**Keywords:** different steel slag content; SMA-13 asphalt mixture; pavement performance; optimal steel slag content

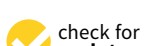



## 1. Introduction

With the rapid development of economic globalization, the level of each industry has also increased, and the consumption of steel materials in economic construction is particularly prominent. According to statistics, world crude steel production reached 1808.6 million tons in 2018, with China accounting for more than 50%. Steel slag is a by-product of the steelmaking process, accounting for 13% of the scrap produced in steelmaking [1]. The utilization rate of steel slag in the United States, Europe, and other developed countries is as high as 70–80% [2]. However, the utilization rate of steel slag resources in China is only 29.5%, of which 50% is used for recycling in road works, sintering, iron-making, and steel-making plants. The remaining steel slag resources are disposed of in large-scale open piles or directly in landfills, which not only occupy land resources but also cause severe pollution to the environment [3]. This is contrary to the sustainable

development goal that many countries are striving to achieve, so the question of how to effectively use steel slag remains [4,5].

Steel slag has better properties, such as roughness and crush resistance, than limestone and basalt [6,7]. Steel slag has abundant metal oxides with rough surfaces due to their aggregation, and the rugged texture of these oxides effectively protects the pyroxene crystals located between them from abrasion and provides a rough and undulating microscopic morphology to the slag surface, causing greater surface friction between steel slag [6].

In recent years, the incorporation of steel slag aggregates into asphalt mixtures has greatly attracted road researchers' interest. The microstructure and surface characteristics of steel slag were analyzed by X-ray diffraction and scanning electron microscopy. The results show that steel slag aggregate has many tiny pores on its surface, which can efficiently adsorb asphalt binder material [7], and the resulting asphalt pavement has higher stability and skid resistance [8–11]. It has been shown that the use of steel slag as coarse aggregate in asphalt concrete pavements can improve the performance of the pavement, especially in terms of durability and stability [1,12–14]. Liapis et al. [9,15] compared paved steel slag asphalt mix sections with conventional limestone asphalt mix pavements and found that the surface macrostructure, as well as the skid resistance of the tested sections, were better. Arbani et al. conducted Marshall tests, dynamic creep tests, and indirect tensile tests on asphalt mixtures incorporated with steel slag to evaluate its mechanical properties and deformation resistance. The results showed that the incorporation of steel slag can effectively improve the Marshall stability and fatigue resistance of asphalt mixes and delay the time of permanent deformation of the mixes. The shortcoming is that the low-temperature crack resistance of steel slag asphalt mixes is lagging [16]. Some researchers have considered grinding steel slag into fine particles and adding it to the hot-mix asphalt mixture in the form of fine aggregates. Test results showed that the steel slag asphalt mixture enhanced the rutting resistance, but the amount of asphalt was significantly increased because of the use of fine steel slag aggregates [5,17]. Kavussi et al. [18] demonstrated the better fatigue resistance of steel slag asphalt mixes by doing four-point bending fatigue tests on steel slag mixes. Chen et al. [19] prepared asphalt mixes by blending gneiss and steel slag aggregates, and they studied the water stability of asphalt mixes by high-temperature damage and low-temperature damage modes. The results indicated that the incorporation of steel slag in asphalt mixes can prevent water damage. Masoudi et al. [20] found that adding steel slag to warm asphalt mixes for aging tests can slow down their short-term and long-term aging. Behnood et al. [21] added steel slag to Stone Mastic Asphalt (SMA) mixes and concluded that the elastic properties of steel slag asphalt mixes were better through performance tests such as Marshall stability and rebound modulus. Phan et al. [22] used infrared camera and microwave heating techniques to analyze the self-healing properties of steel slag asphalt mixes. The results showed that the best ductility and crack self-healing properties were achieved when 30% of average coarse aggregate was replaced by steel slag. However, there are many irregular protrusions and tiny pores on the surface of steel slag aggregates, which cause asphalt mixes to be difficult to be compacted and have a high void ratio [23], and the pores can absorb large amounts of asphalt, resulting in increased asphalt usage [5]. Another problem is that the chemical composition of the steel slag aggregate contains substances such as free f-CaO and f-MgO that react with $CO_2$ in the air, mainly to form $CaxMg_1$-x$CO_3$. If these components are hydrated, they can cause pavement cracking [24]. Therefore, steel slag aggregates should be placed in an open environment or immersed in water for at least six months to allow them to fully react before they are recycled and then made into asphalt mixes [25] for better results. Research has shown that a short-term rutting cycle number of 2520 cycles can simulate heavy traffic volumes, predict long-term rutting, and also reflect the performance of asphalt mixtures; however, the A/C index, complex stability index, and shear index were proven to be reliable indicators to verify the performance of the recycled asphalt mix [26]. Researchers evaluated the effect on rutting resistance and fatigue cracking using

the locking point concept of aggregates through performance tests of asphalt mixtures, and the concept of interlocking points during compaction of the mix was verified [27].

In summary, mixing steel slag aggregates into asphalt mixes can improve their skid resistance, stability, water damage resistance, self-healing, and rutting resistance. However, there are few studies on the optimal amount of steel slag incorporation for SMA-13 asphalt mixes. In this paper, SMA-13 asphalt mixes prepared with different amounts of steel slag aggregates instead of common limestone coarse aggregates were used and analyzed for road performance and volume expansion. It was concluded that an SMA-13 asphalt mixture with a certain amount of steel slag doping had the best road performance, which provides the theoretical basis for later application and development in practical engineering.

The objectives of this study are: to evaluate the mechanical properties of five types of asphalt mixes with steel slag by means of wheel tracking, Hamburg wheel tracking (HWT), three-point bending, freeze–thaw splitting, dynamic modulus, and uniaxial penetration tests. The purpose is to determine the optimum amount of steel slag for SMA-13 asphalt mixes to provide a theoretical basis for better application to road projects.

## 2. Materials and Methods

### 2.1. Materials

The asphalt binder used in this paper was SBS-modified asphalt produced by JingBo Petrochemical Company (Binzhou, China). According to the Chinese standard JTG E20-2011 [28], the conventional properties of the asphalt were tested, and the specific technical indices are shown in Table 1, which all meet the requirements of certain specifications.

**Table 1.** Technical index of SBS modified asphalt.

| Items | Test Values | Specification [29] |
|---|---|---|
| Penetration (25 °C, 0.1 mm) | 70.1 | 60–80 |
| Softening point (°C) | 64.5 | $\geq$55 |
| Flash point (°C) | 272 | $\geq$230 |
| Ductility (5 °C, cm) | 46.3 | $\geq$30 |

The steel slag was made of hot-sealing steel slag aggregate produced by the Rizhao Iron and Steel Plant in Rizhao, China. To prevent the volume expansion caused by the reaction of steel slag with water, it was placed in a natural environment while exposed to rain and air for eight months, which basically eliminated volume instability. The limestone used was high-quality limestone from Jinan, China. Based on the Chinese standard JTG E42-2005 [30], each functional index of coarse aggregate and fine aggregate was tested, and the specific indices are shown in Tables 2–5, which met the specification requirements. If steel slag fine aggregate is used for an asphalt mixture, it may cause a larger volume change and an additional increase in the amount of asphalt used [31]. Therefore, in this study, fine limestone aggregate was used as the fine aggregate.

**Table 2.** Properties of coarse aggregates.

| Items | Unit | Steel Slag Test Results | Limestone Test Results | Specification [29] |
|---|---|---|---|---|
| Apparent relative density | g/cm$^3$ | 3.543 | 2.726 | $\geq$2.6 |
| Water absorption | % | 1.930 | 0.556 | $\leq$2.0 |
| Crush value | % | 9.60 | 19.8 | $\leq$26 |
| Abrasion value | % | 11.1 | 22.3 | $\leq$28 |
| Soundness | % | 2.7 | 6.0 | $\leq$12 |

**Table 3.** Properties of fine aggregates.

| Items | Unit | Test Values | Specification [29] |
|---|---|---|---|
| Apparent relative density | g/cm$^3$ | 2.725 | $\geq$2.5 |
| Sand equivalent | % | 73 | $\geq$60 |
| Soundness | % | 15 | $\geq$12 |
| Angularity | s | 46 | $\geq$30 |

**Table 4.** Properties of fillers.

| Items | Unit | Test Values | Specification [29] |
|---|---|---|---|
| Apparent relative density | t/m$^3$ | 2.700 | $\geq$2.50 |
| Water content | % | 0.1 | $\leq$1 |
| Appearance | - | No agglomerates | No agglomerates |
| Hydrophilic coefficient | - | 0.49 | $<$1 |
| Plasticity index | % | 2.2 | $<$1 |

**Table 5.** Steel slag SMA-13 gradation composition design.

| | Steel Slag Content 0% | Steel Slag Content 25% | Steel Slag Content 50% | Steel Slag Content 75% | Steel Slag Content 100% |
|---|---|---|---|---|---|
| Steel slag 10–15 mm/% | 0 | 11 | 22 | 32 | 40 |
| Limestone 10–15 mm/% | 38 | 28 | 18 | 8 | 0 |
| Steel slag 5–10 mm/% | 0 | 11 | 22 | 31 | 40 |
| Limestone 5–10 mm/% | 39 | 28 | 17 | 9 | 0 |
| Limestone 0–3 mm/% | 13 | 12 | 12 | 11 | 11 |
| mineral powder/% | 10 | 10 | 9 | 9 | 9 |
| optimum asphalt content/% | 5.14% | 5.25% | 5.34% | 5.50% | 5.61% |

The mineral powder used in this study was made from ground limestone, acting as a filler in the asphalt mix with the aim of reducing the voids in the asphalt concrete. The mineral powder and asphalt together form an asphalt mastic, which improved the strength and stability of the asphalt concrete; the specific indicators are shown in Table 4.

To determine the optimal steel slag incorporation for the SMA-13 asphalt mixes, a total of five mix gradations were designed with different steel slag incorporation levels [32]: 0%, 25%, 50%, 75%, and 100% steel slag content. The coarse aggregate part of the SMA-13 asphalt mixture was formed by combining the steel slag and limestone with particle sizes of 5–10 mm and 10–15 mm at certain ratios. The fine aggregate part was limestone with a particle size of 0–3 mm. The volumetric method [33,34] was used to replace the limestone coarse aggregates with steel slag coarse aggregates at different dosing levels while preventing excessive density differences between the two, resulting in deviations between the actual synthetic gradation curve and the target gradation curve. The Marshall compaction test was used to determine the best amount of asphalt. Road performance verification was performed, and the specific grade composition is shown in Table 5.

### *2.2. Experimental Methods*

Performance verification of the SMA-13 asphalt mixes was conducted using the wheel tracking, HWT, three-point bending, freeze–thaw splitting, dynamic modulus, uniaxial penetration, and asphalt mix expansion tests, as shown in Figure 1.

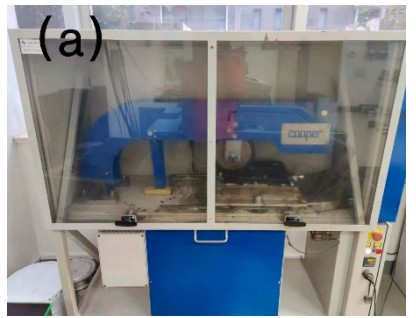
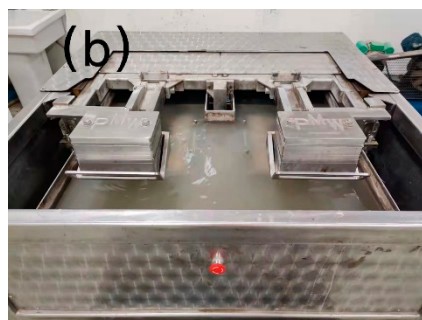
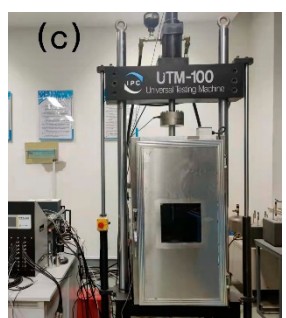
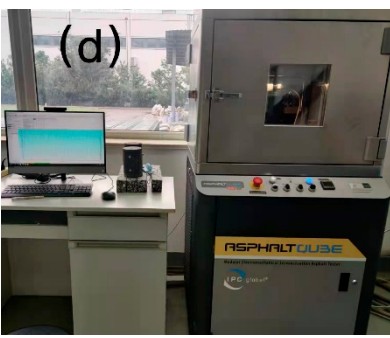
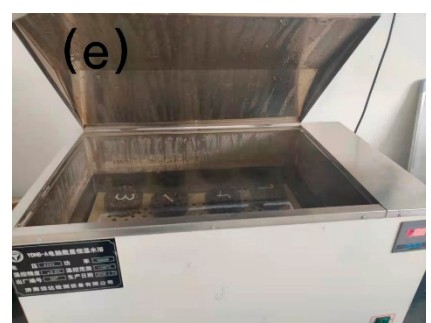
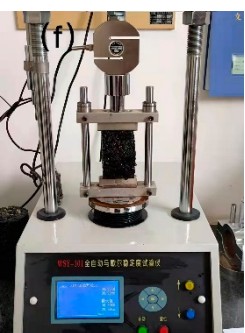

**Figure 1.** Experimental images of asphalt mixture tests used in this research: (**a**) wheel tracking test, (**b**) Hamburg wheel tracking test, (**c**) three-point bending test, (**d**) dynamic modulus test, (**e**) asphalt mix expansion test, and (**f**) freeze–thaw splitting test.

### 2.2.1. Wheel Tracking Test

The incorporation of steel slag into SMA-13 asphalt mixes usually improves their rutting resistance due to the strength of the slag itself [35]. According to the Chinese standard JTG E20-2011 (T0719) [28], the rutting resistance of asphalt mixture is evaluated by the wheel tracking test, and its dynamic stability can better reflect the ability of asphalt pavement to resist rutting formation under high-temperature conditions in summer. The test used standard asphalt mixture specimens with a length of 300 mm, width of 300 mm, and height of 50 mm. Before the test began, the mixture specimen was placed in a thermostat at $60 \pm 1\ °C$ for 6 h to ensure that the internal temperature was stable at $60\ °C$. Then, the specimen was placed at the test temperature of $60\ °C$, the contact pressure between a solid tire made of rubber and the specimen was 0.7 MPa, and the round-trip crimping speed was $42 \pm 1$ time/min. A linear variable differential transformer (LVDT) collected the rut depth change and calculated the dynamic stability based on the change of time and rut depth, as shown in Equation (1):

$$DS = \frac{(t_2 - t_1) \times N}{d_2 - d_1}, \tag{1}$$

where *DS* is the dynamic stability (cycles/mm); *N* is the loading peed (42 cycles/min); $d_1$ is the amount of deformation corresponding to time $t_1$ (mm); and $d_2$ is the amount of deformation corresponding to time $t_2$ (mm).

### 2.2.2. Hamburg Wheel Tracking (HWT) Test

The HWT test is considered in the literature [36] as a reproducible and reliable method for characterizing the premature failure of asphalt mixtures. According to the standard JTG E20-2011 [28], the HWT test can evaluate the rutting resistance and water stability of asphalt mixtures. Cylindrical specimens of asphalt mixture with a height of 65 mm and a diameter of 150 mm formed by a rotary compactor were first cut into Hamburg standard specimens using a cutting machine [37]. Then, we placed a mold where the test piece was installed into 50 $\pm$ 1 °C water. A steel wheel was made to perform roller compaction on the specimen at a rate of 52 $\pm$ 2 times per minute. When the steel wheel crushed the specimen 20,000 times or the rutting depth measured by the LVDT reached 12.5 mm, the rutting meter automatically stopped and saved the data. This test was mainly used to determine the early damage of the asphalt mixture by measuring the rutting depth and loading times.

### 2.2.3. Freeze–Thaw Splitting Test

The freeze–thaw splitting test is based on the standard JTG E20-2011 [28]. It evaluates the resistance of asphalt mixtures to water damage by the splitting tensile strength ratio of asphalt mixture specimens before and after freezing and thawing [38]. A standard specimen undergoes Marshall compaction 50 times on each side, and the specimen size is 100 mm in diameter and 63.5 $\pm$ 2.5 mm in height. The specimens were divided into two groups; one group was placed in a water bath at 25 °C for 2 h $\pm$ 10 min before the test, while another group was first placed in a vacuum. The degree of saturation was at 70–80% before placing the test specimen into a plastic bag filled with 10 mL of water. Then, the plastic bag was placed in a thermostat at –18 °C for at least 16 h, chilled, and then placed in a water bath at 60 °C for 24 h. Then, the specimens were moved to a water bath at 25 °C for 2 h, and the splitting strength of the Marshall specimens was measured according to a loading rate of 50 mm/min.

### 2.2.4. Dynamic Modulus Test

The dynamic modulus test applies an offset sine wave axial compressive stress to the specimen according to a specific temperature and loading frequency to measure the recoverable strain of the specimen, which is an important method to simulate the actual road performance of asphalt mixtures. According to the Chinese standard JTG E20-2011 (T0738) [28], cylindrical specimens of 150 mm in diameter and 170 mm in height formed by a rotary compaction apparatus were cut into standard specimens of 100 mm in diameter and 150 mm in height by coring. The specimens were held at the test temperature for at least 4 h before the test. The AST material testing machine was used for testing. The dynamic modulus can be calculated by Equation (2):

$$|E^*| = \frac{\sigma_0}{\varepsilon_0},\tag{2}$$

where $|E^*|$ is the dynamic modulus (MPa), $\sigma_0$ is the applied maximum stress (MPa), and $\varepsilon_0$ is the measured peak strain.

### 2.2.5. Three-Point Bending Test

The three-point bending test was used to evaluate the performance of asphalt mixtures against bending and tensile damage at low temperatures. According to the Chinese specification JTG E20-2011 (T0715) [28], the asphalt mixture specimen (300 mm $\times$ 300 mm $\times$ 50 mm) was cut into small, prismatic beam specimens with a length of 250 $\pm$ 2 mm, width of 30 $\pm$ 2 mm, and height of 35 $\pm$ 2 mm. Before the test, the specimens were placed in insulation at –10 $\pm$ 0.5 °C for at least 45 min to ensure that its internal temperature was uniform. Then, the specimen was placed on a base with a span diameter of 200 mm, and a load was applied to the middle of the specimen at a loading speed of 50 mm/min. The mid-span deflection was measured using the LVDT. To reduce the occurrence of errors, four replicate tests were used for each asphalt mixture, and the flexural tensile

strength, maximum flexural strain, and bending stiffness modulus were calculated using Equations (3)–(5). The low-temperature durability of the steel slag SMA-13 asphalt mixes was determined by measuring the crack resistance:

$$R_B = \frac{3LP_B}{2bh^2},$$ (3)

$$\varepsilon_B = \frac{6hd}{L^2},$$ (4)

$$S_B = \frac{R_B}{\varepsilon_B},$$ (5)

where $R_B$ is the flexural tensile strength of the specimen at the time of damage (MPa), $\varepsilon_B$ is the maximum bending strain ($\mu\varepsilon$), $S_B$ is the modulus of bending stiffness of the specimen at the time of damage (MPa), $b$ is the width of the span section of the specimen (mm), $h$ is the height of the beam (mm), $L$ is the span of the testing fixture, 200 mm, $P_B$ is the maximum load when the specimen is damaged, and $N$ is the mid-span deflection of the specimen when breaking the ring (mm).

### 2.2.6. Uniaxial Penetration Test

Referring to the Chinese specification JTG D50-2017 [39], a standard cylindrical specimen with a diameter of 150 mm and a height of 100 mm was placed in a constant-temperature chamber at $60 \pm 0.5\,°C$ and insulated for 5–6 h. A UTM-100 tester was used to maintain a loading rate of 1 mm/min to apply a 42 mm diameter loading indenter to the specimen before stopping the test when the stress value dropped to 90% of the extreme value point. To reduce the occurrence of errors, four replicate tests were used for each asphalt mixture. The penetration strength of the specimen is calculated using Equations (6) and (7):

$$R_l = f_l\sigma_P,$$ (6)

$$\sigma_P = \frac{P}{A},$$ (7)

where $R_l$ is the penetration strength (MPa), $\sigma_P$ is the penetration stress (MPa), $P$ is the ultimate load when the specimen is damaged, N, $A$ is the cross-sectional area of the indenter (mm$^2$), and $f_l$ is the penetration stress coefficient. When the diameter of the test piece was 150 mm, $f_l = 0.35$.

### 2.2.7. Asphalt Mix Expansion Test

According to the Chinese specification JTG E42-2005 [30], when steel slag is used as a material for road asphalt layers, its activity and swelling must be tested to see if it meets the use standards. At least three standard Marshall specimens were made for each steel slag content. The diameters and heights of the specimens were measured with vernier calipers at three and four places, respectively, to calculate the initial volume V1. Then, the specimens were placed into a constant-temperature water bath at $60 \pm 1\,°C$ for 72 h before removal and cooling to room temperature. Then, the appearance of cracks or bulging phenomenon were observed, and the new volume of the specimen V2 was measured according to the same method as before. Equation (8) was used to calculate the expansion of the steel slag asphalt mixture:

$$C = \frac{v_2 - v_1}{v_1} \times 100,$$ (8)

where $C$ is the expansion amount of the steel slag asphalt mixture (%), $v_1$ is the volume of the specimen before the water bath (cm$^3$), and $v_2$ is the volume of the specimen after the water bath (cm$^3$).

## 3. Results

### 3.1. Study on High-Temperature Stability

The high-temperature rutting resistance of the SMA-13 asphalt mixes with different steel slag contents was analyzed by the dynamic stability and rutting depth obtained from the wheel tracking and HWT tests with the incorporation of steel slag. Figure 2 shows the variations in dynamic stability and rutting of the asphalt mixes at high-temperature conditions with the increase in steel slag content. The test results showed that the dynamic stability of the mixes increased with the increase in steel slag content, and all samples had a stability that was greater than the specification value of 3000 times/mm [29]. The dynamic stability reached a peak of 8000 times/mm when the slag content was 75% before declining. However, the SMA-13 asphalt mixes with 25%, 50%, 75%, and 100% steel slag were all at least 25% more dynamically stable than those without steel slag. The rutting depth of the SMA-13 asphalt mixture with 75% steel slag was 1.19 mm, which was smaller than the other samples. Figure 3 shows that the rutting depth of the HWT test conducted at 50 °C in a water bath decreased and then increased with the increase in steel slag content, reaching a minimum value of 2.25 mm when the steel slag content reached 75%. The addition of steel slag can improve the high-temperature rutting resistance of SMA-13 asphalt mixes because steel slag is harder and more angular than limestone, and it has a stronger ability to resist pressure. The compacted coarse aggregate can form a tightly embedded locking structure, and steel slag is alkaline, has more internal pores, and bonds more efficiently with asphalt, thus improving the high-temperature stability of the asphalt mixture. However, as the steel slag content increased to 100%, the amount of asphalt required increased, and the asphalt inside the pores of the slag reached saturation, which is more likely to lead to flooding oil and rutting under the same compaction conditions. As the amount of steel slag content increases, the porosity and compaction may be more difficult to control due to the unique angularity of the slag leading to a more difficult mix structure to be compacted, resulting in reduced high-temperature stability.

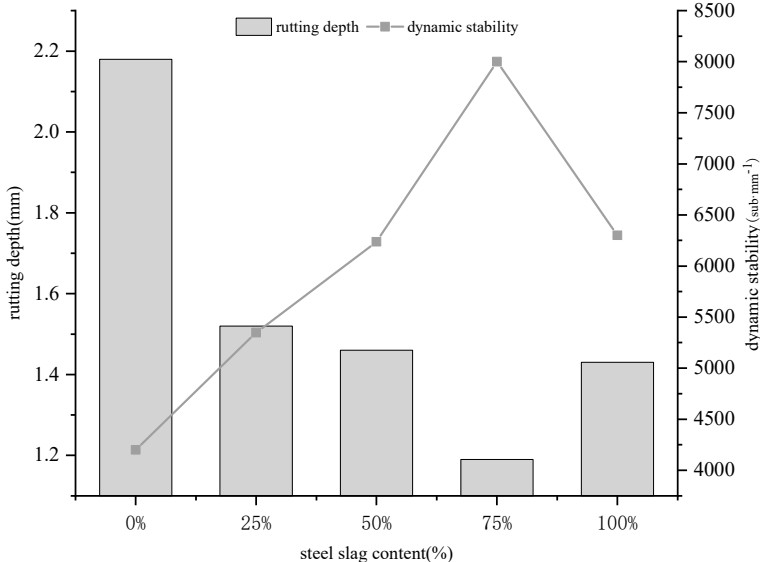

**Figure 2.** Wheel tracking test results of asphalt mix with different steel slag contents.

### 3.2. Study on Low-Temperature Crack Resistance

The maximum flexural tensile strength and maximum flexural strain of SMA-13 asphalt mixes were analyzed according to the Chinese standard JTG D50-2017 [39] to evaluate their low-temperature crack resistance. Asphalt mixes with good low-temperature crack resistance can be characterized by maximum bending and tensile strains. According to Table 6, the maximum bending and tensile strains of the samples were better than the

specifications [29]. However, the maximum bending and tensile strains of the SMA-13 asphalt mixture with 0% steel slag content were the largest. These values started to decrease with the increase in steel slag content, and they decreased by 5.98% when the steel slag content was 75%. Mainly because of the steel slag in the open storage process, its surface will accumulate many tiny dust particles adsorbed in the pores of the steel slag, reducing the adhesion with asphalt. With the increase in steel slag, the amount of asphalt increases, but the above phenomenon will be more obvious. Therefore, with the increase in steel slag content, the low-temperature crack resistance of the SMA-13 asphalt mixture will decrease. However, when comparing a steel slag content of 75% and 0%, the low-temperature crack resistance did not decrease much.

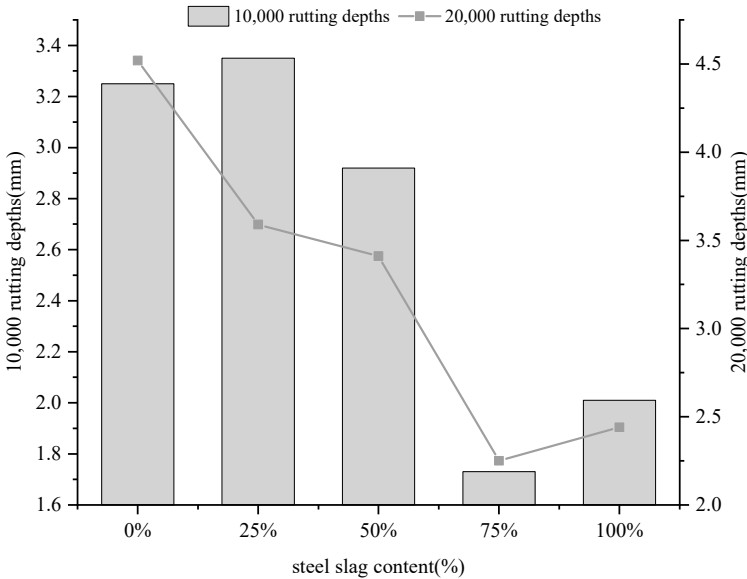

**Figure 3.** Hamburg wheel tracking test results of asphalt mix with varying steel slag content.

**Table 6.** Results of low-temperature bending tests on SMA-13 asphalt mixes with varying steel slag content.

| Steel Slag Content (%) | Flexural Tensile Strength (MPa) | Maximum Bending Strength (με) | Bending Stiffness Modulus (MPa) |
| --- | --- | --- | --- |
| 0 | 16.72 | 3091.74 | 4402.20 |
| 25 | 15.04 | 2979.04 | 4563.85 |
| 50 | 14.29 | 2925.42 | 4883.31 |
| 75 | 13.27 | 2906.60 | 5048.87 |
| 100 | 12.55 | 2852.06 | 5408.72 |

*3.3. Study on Water Stability*

The test conditions of the freeze–thaw splitting test were more stringent than the general water immersion test. The purpose was to test the resistance of the asphalt mixes to water damage while varying the steel slag content. From Figure 4, it can be seen that the freeze–thaw splitting strength ratio of the asphalt mix gradually increased with the incorporation of steel slag, but the increase was not significant. The freeze–thaw splitting strength ratio of the asphalt mixture reached a maximum of 91.5% when the steel slag content reached 75%, which was 7.39% higher than that of 0% slag, followed by a 3.5% decrease in the freeze–thaw splitting strength ratio when the steel slag content was 100%. This indicated that the incorporation of steel slag can improve the water stability of SMA-13 asphalt mixes. The main reason for this is that there are many tiny pores inside the steel slag, which is also alkaline, and the asphalt combined more densely, Thus, when combined with asphalt, the steel slag can increase the adhesion force and improve the water stability

of the mixture. However, there are harmful impurities in the steel slag such as CaO and MgO that have not fully reacted, and a reaction with water will generate $Ca(OH)_2$ and $Mg(OH)_2$, which affects the water stability of the steel slag SMA-13 asphalt mixture.

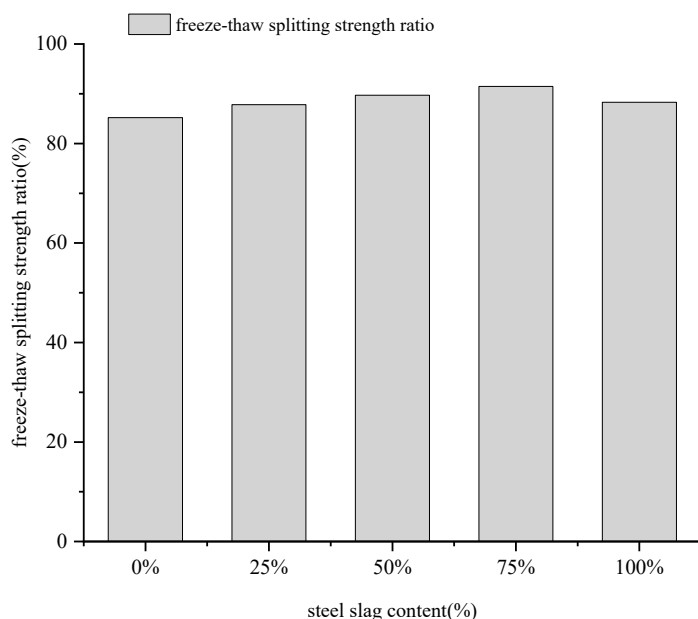

**Figure 4.** Freeze–thaw splitting strength ratio of SMA-13 under varying steel slag content.

*3.4. Study on Dynamic Modulus*

The main reason for using the uniaxial compression dynamic modulus as an essential performance parameter for asphalt mixture materials is that it is closer to the actual material response characteristics of pavement and to the assumption of an elastic layered system. Figure 5 represents the increase in the dynamic modulus of the asphalt mixture with the increase in loading frequency at a constant test temperature. As the material properties of asphalt mixture are viscoelastic, there is a certain delay in its deformation under the external load stress. Part of the transient energy release is not sufficient, and the accumulation of energy with the increase in loading frequency gradually increases, which also leads to a gradual increase in the dynamic modulus $|E^*|$. The dynamic modulus decreases with increasing temperature when the loading frequency is the same, as was seen in the SMA-13 asphalt mixtures. This showed that the increase in temperature affected the dynamic modulus of steel slag SMA-13 asphalt mixes because the adhesion between the internal pores of steel slag and asphalt decreased when the temperature increased, and the incorporation of steel slag will be accompanied by a higher asphalt dosage. The above phenomenon will be more obvious. According to the Chinese specification JTG D50-2017 [39], the dynamic compression modulus of the designed SMA-13 asphalt mixture measured at 20 °C and 10 Hz needs to fall in the range of 7500–12,000 MPa. Therefore, as can be seen from Figure 6, a steel slag content of 0% and 25% did not meet the specification requirements. However, the dynamic modulus measured at 50%, 75%, and 100% steel slag content met the specification requirements. Therefore, incorporating an appropriate amount of steel slag into SMA-13 asphalt mixes can improve the dynamic modulus values. In summary, it can be seen that the dynamic modulus when the steel slag content was 50%, 75%, and 100% met the specification requirements, where the dynamic modulus was larger at 75% and 100%. However, the dynamic modulus decreased more rapidly with the increase in temperature when the steel slag content was 100%. Thus, the recommended amount of steel slag content for SMA-13 asphalt mixes is 75%.

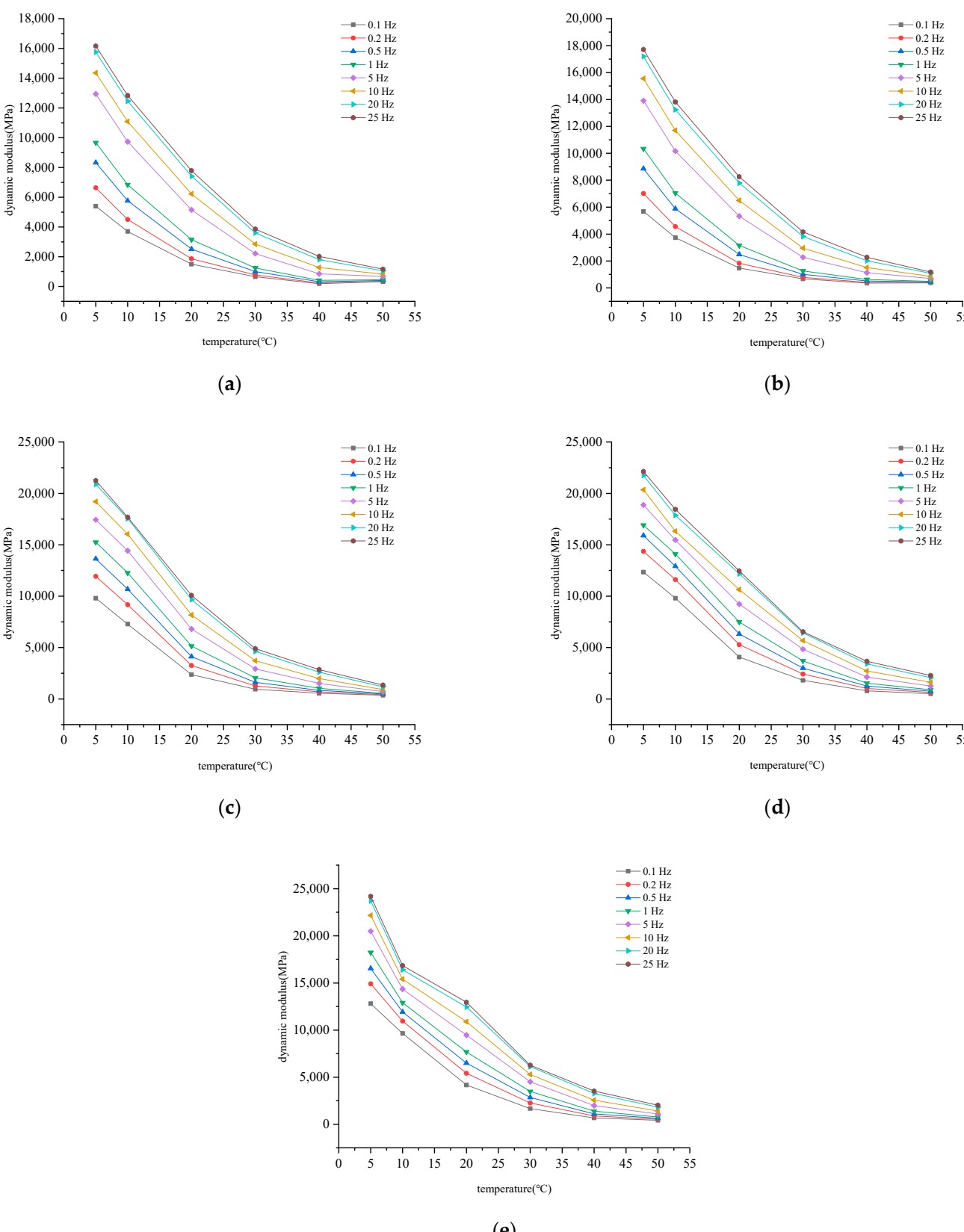

**Figure 5.** Dynamic modulus test results of asphalt mixes with varying steel slag content: (**a**) 0%; (**b**) 25%; (**c**) 50%; (**d**) 75%; and (**e**) 100% steel slag content.

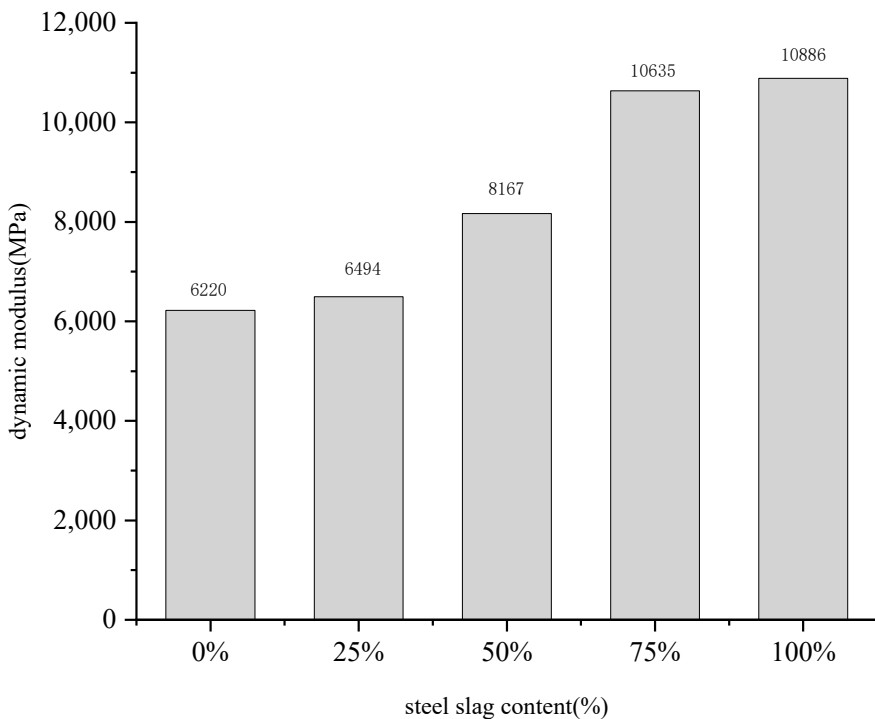

**Figure 6.** Experimental results on the relationship between dynamic modulus and steel slag content at 20 °C and 10 Hz.

### 3.5. Study on Shear Performance

Based on the penetration strength calculated from the uniaxial penetration test, the change in shear resistance of the SMA-13 asphalt mixture was analyzed for different amounts of steel slag. The test results are shown in Figure 7. With the increase in steel slag content, the penetration strength of the SMA-13 asphalt mixture first increased and then gradually decreased. The penetration strength of the asphalt mixture reached its peak when the steel slag was mixed at 75%. Steel slag is an alkaline material, and asphalt easily reacts with alkaline materials to form chemical bonds. As a result, the contribution to the shear strength of the mixture is greater, and as the proportion of coarse aggregate in SMA-13 asphalt mixes is increasingly replaced by steel slag, the cohesion increases. Steel slag has irregular angularity, so the inlaid friction force and sliding friction force between steel slag and steel slag is larger than that of limestone aggregates. In addition, the strong locking structure formed by the asphalt mixture after grinding has a greater gain on the internal friction angle; however, when the amount of steel slag exceeds a certain percentage, it will cause the asphalt mixture to be less easily compacted. Furthermore, the sliding friction force between the aggregates will be relatively increased, resulting in a decrease in the internal friction angle. Comprehensive analysis of the shear resistance of the mixture is better when the amount of steel slag is 75%.

### 3.6. Study on Volume Stability

The contents of f-CaO and f-MgO present in steel slag will increase with the increase in steel slag content, and they will swell to different degrees when mixed with water, which will lead to the appearance of cracks and other diseases in the asphalt mixture [5] and reduce the performance of the road. The asphalt mixture needs to be tested for expansion to check whether its volume expansion rate meets the specification requirements. According to the requirements of Chinese specification JTG E42-2005 [30], the volume change before and after the test is not to be more than 1.5%. The test results are shown in Table 7. The volumetric expansion of the SMA-13 asphalt mixture increased gradually with the increase in steel slag content. The main reason is that with the increase in steel slag admixture,

the content of f-CaO and f-MgO inside the slag also increased and reacted with water to produce $Ca(OH)_2$ and $Mg(OH)_2$ [38], which led to volume expansion and affected the volumetric stability of the asphalt mixture. The volume expansion rate of the SMA-13 asphalt mixtures with varying steel slag content were all less than 0.9%, which met the specification requirements. No cracks on the surface or other undesirable diseases appeared. It may be that the treatment with the steel slag reduced the content of f-CaO and f-MgO more effectively, and asphalt wrapped around the surface of the steel slag, reducing the chance of contact with moisture, thus reducing the occurrence of swelling. The volumetric stability of the steel slag SMA-13 asphalt mixture is inversely proportional to the amount of steel slag contained in the mixture.

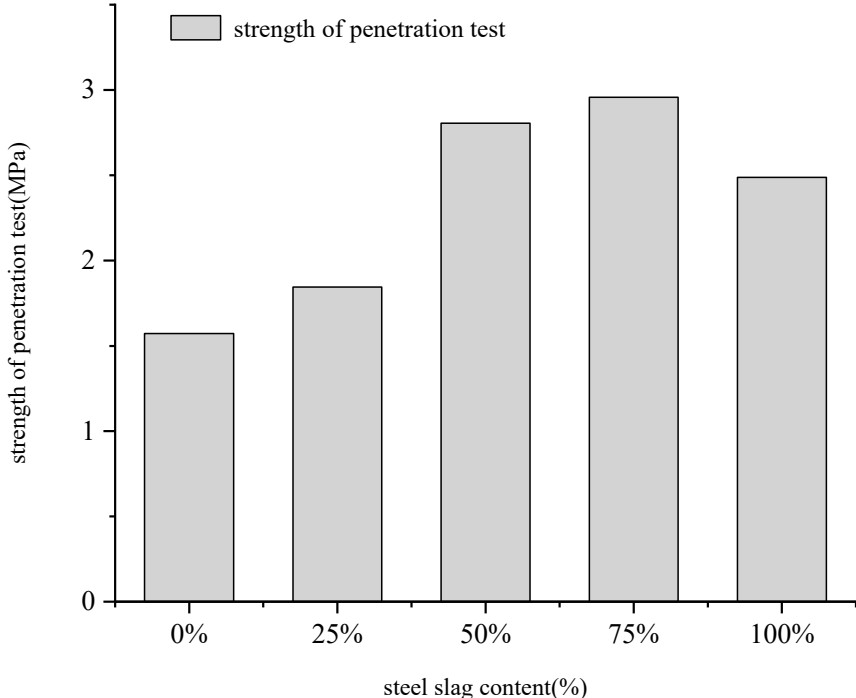

**Figure 7.** Test results of penetration strength of SMA-13 asphalt mix with different steel slag contents.

**Table 7.** Results of swelling tests on SMA-13 asphalt mixes with different steel slag contents.

| | Volume before Immersion (cm³) | Volume after Immersion (cm³) | Volume Expansion Ratio (%) | Surface State of Specimen after Immersion |
|---|---|---|---|---|
| Steel slag content 0% | 511.069 | 512.352 | 0.251 | |
| Steel slag content 25% | 512.123 | 513.931 | 0.353 | No cracks on the surface, no obvious drumming |
| Steel slag content 50% | 508.177 | 510.347 | 0.427 | |
| Steel slag content 75% | 508.052 | 511.593 | 0.697 | |
| Steel slag content 100% | 523.717 | 528.200 | 0.856 | |

## 4. Conclusions

In this study, the limestone coarse aggregate in SMA-13 asphalt mixes was replaced with steel slag. Five SMA-13 asphalt mixes prepared with varying steel slag content (0%, 25%, 50%, 75%, and 100%) were tested by the wheel tracking, Hamburg wheel tracking, three-point bending, freeze–thaw splitting, dynamic modulus, uniaxial penetration, and asphalt mix expansion tests to investigate the effect of increasing steel slag content on performance. Based on the results of the various road performance tests, the following conclusions can be drawn.

(1) Compared to the steel slag content of 0%, the dynamic stability of SMA-13 asphalt mixes with a steel slag content of 25%, 50%, 75%, and 100% increased by 27.33%, 48.5%, 90.48%, and 50.00%, respectively. When the steel slag content was 75%, the rutting depths for both the wheel tracking test and the Hamburg wheel tracking test reached minimum values of 1.19 mm and 2.25 mm, respectively. Demonstrating that the incorporation of steel slag can improve the high-temperature stability of SMA-13 asphalt mixes, the best improvement was achieved when the steel slag content was 75%. However, an increased steel slag content can lead to a reduction in the low-temperature crack resistance of the asphalt mix.

(2) The water stability of the asphalt mixes tended to increase and then decrease with the incorporation of steel slag. The freeze–thaw splitting ratio was 91.5%, and the water stability was optimal when the steel slag content was 75%. The presence of hazardous substances in steel slag can lead to poor volumetric stability of the asphalt mix, which decreases with increasing amounts of steel slag.

(3) The dynamic modulus of the asphalt mix increased as the steel slag content increased. At a temperature of 20 °C and a loading frequency of 10 Hz, the dynamic modulus of the SMA-13 asphalt mixes with 75% steel slag content increased by 70.98% compared to those with 0% steel slag content. The shear resistance of the SMA-13 asphalt mixes was strongly influenced by the embedded effect between the steel slag aggregates. As the slag content increased, the shear resistance tended to increase and then decrease. When the slag content was 75%, the penetration strength of the mix reached a maximum of 2.957 Mpa with an optimal shear resistance.

By analyzing the road performance of SMA-13 asphalt mixes with varying steel slag content, the optimum content was finally determined to be 75%.

**Author Contributions:** W.C.: Writing—original draft; X.Z., W.C.: Data curation; X.X., W.H., Z.L. and G.H.: Methodology; W.H., X.X., Z.L. and G.H.: Project administration; X.Y., J.W. and X.Z.: investigation; X.Y. and J.W.: Supervision. All authors have read and agreed to the published version of the manuscript.

**Funding:** This research was funded by National Key R&D Program of China, grant number (2018YFB1600100), Shandong Natural Science Foundation Committee (ZR2020QE271, ZR2020KE024) and Shandong Province Key R&D Program (2019GSF109020, 2019GGX101042).

**Institutional Review Board Statement:** Not applicable.

**Informed Consent Statement:** Not applicable.

**Data Availability Statement:** Data sharing is not applicable to this article.

**Acknowledgments:** The authors would like to thank the Shandong Transportation Institute for their support.

**Conflicts of Interest:** The authors declare no conflict of interest.

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
