# Peer review of "Study on the Optimum Steel Slag Content of SMA-13 Asphalt Mixes Based on Road Performance"

_coatings, doi:10.3390/coatings11121436_

Round 1

Reviewer 1 Report

This paper presents evaluation the optimum steel slag content of SMA-13 asphalt mixes based on road performance. For this purpose, five types of SMA-13 asphalt mixes were designed and prepared by replacing coarse limestone aggregate with steel slag of equal volume. Then, the optimum dosing of steel slag was analyzed. Comparisons between the results shows that the incorporation of steel slag increases the dynamic modulus of the asphalt mix.This is an interesting paper. However, this reviewer does not recommend the publication of the manuscript in the present form because of the following reasons:

1) This reviewer think it will be useful if the authors provide some additional information on the Materials and Methods used in this study.

2) English usage and spelling should be improved.

3) The manuscript is focused on properties of flexible pavements and needs better description of the properties of asphaltic materials in pavement design, such as linear and nonlinear viscoelastic properties.  They may use available literature such as the following reference:

  • (2020). Effect of Evotherm-M1 on Properties of Asphaltic Materials Used at NAPMRC Testing Facility. Journal of Testing and Evaluation48(3).

This paper will recommend for publication if the authors consider the above suggestions to improve the quality of the manuscript. Some editing is still needed. I believe it will be done before publishing.

Reviewer 2 Report

This manuscript covers a laboratory study on the optimum steel slag content of SMA-13 asphalt mixes based on road performance. The effect of incorporating of steel slag in SMA-13 asphalt mixes on properties such as temperature stability, water stability, shear resistance, crack resistance, volumetric stability and dynamic modulus has been evaluated. The manuscript has been carefully reviewed and has been found to be well written requiring a few corrections. Find below for your consideration comments with request for clarifications and/ or further improvement of the manuscript:

  1. Provide references for the standards and test methods presented in the manuscript e.g., JTG E20-2011, JTG E42-2005, JTG E20-2011 (T0719), JTG E20-2011 (T0738), JTG E20-2011 (T0715), JTG D50-2017, JTG E42-2005, AASHTO T 283 etc.
  2. Provide references for the specification’s requirements provided in Tables 1-4 in the manuscript.
  3. In Line 140, what is the main difference between Wheel tracking test and Hamburg Wheel tracking test? Why do you need both? Is there a specific parameter which one of the tests measures which the other does not measure?
  4. In Line 162, , it is stated “N is the loading peed”. It should be speed instead of “peed”.
  5. In Line 198, it is stated “AST material testing machine”. What is AST in full?
  6. In Line 275–276, it is stated “specification value of 3000 times/mm”. Provide references for the specification. Same applies in other parts of the manuscript where there is reference to specifications.
  7. In Line 292, it is stated “flooding oil”. What does this mean?
  8. In Line 308–310, it is stated that “According to Table 6, the maximum bending and tensile strains of SMA-13 asphalt mixture with different steel slag content are more significant than the specification” What is the specification and value?
  9. In Line 337–339, it is stated that “However, there are harmful impurities in the steel slag such as CaO and MgO that have not fully reacted, and the reaction with water will generate Ca(OH)2 and Mg(OH)2 to affect the water stability of the steel slag SMA-13 asphalt mixture” However, in In Line 111– 112, it is stated that“the steel slag is placed in a natural environment with rain and air in full contact for eight months”. Explain whether this is not relevant in relation to mitigating harmful impurities in the steel slag such as CaO and MgO.
  10. The conclusion section must be rewritten to clearly bring out the key findings of the study. For example, the first conclusion is simply stating volume replacement method used in the study without any key finding being presented.

Reviewer 3 Report

  • Your abstract does not tell the story. It looks like a bunch of sentences put together. Improve the flow. Start with a short introduction, then an objective, and some major conclusions.
  • Your introduction needs more information about performance. I missing some general information regarding the performance of asphalt mixtures such as https://doi.org/10.1016/j.conbuildmat.2017.07.164 and doi.org/10.1080/14680629.2021.1908408.
  • The paper needs a lot of language improvement.
  • You don`t have objectives.
  • Provide mix design data (volumetrics).

Overall, the manuscript looks like some technical report. You need to provide some scientific findings, not just to compare some data. State clearly your contributions to scientific knowlage.

Round 2

Reviewer 1 Report

Did not address my previous comment. Please recheck. 

Reviewer 3 Report

Thank you for addressing my comments.